# Plasmonic and Photothermal Effects of CuS Nanoparticles Biosynthesized from Acid Mine Drainage with Potential Drug Delivery Applications

**DOI:** 10.3390/ijms242216489

**Published:** 2023-11-18

**Authors:** Hernán Escobar-Sánchez, Claudio Carril Pardo, Noelia Benito, Jacobo Hernández-Montelongo, Iván Nancucheo, Gonzalo Recio-Sánchez

**Affiliations:** 1Departamento de Física, Universidad de Concepción, Concepción 4070386, Chile; hescobar2016@udec.cl (H.E.-S.); noelia.benito@udec.cl (N.B.); 2Facultad de Ciencias de la Salud, Universidad San Sebastián, Concepción 4080871, Chile; claudio.carril@uss.cl; 3Departamento de Ciencias Matemáticas y Físicas, Universidad Católica de Temuco, Temuco 4823302, Chile; jacobo.hernandez@uct.cl; 4Facultad de Ingeniería, Arquitectura y Diseño, Universidad San Sebastián, Concepción 4080871, Chile; ivan.nancucheo@uss.cl

**Keywords:** CuS nanoparticles, acid mine drainage, plasmon resonance, photothermal, drug delivery

## Abstract

In this work, the plasmonic and photothermal effects of CuS nanoparticles biosynthesized from acid mine drainage (AMD) were studied. CuS were formed by delivering the H_2_S generated by a sulfidogenic bioreactor to an off-line system containing the AMD. The precipitates collected after contact for an hour were washed and physico-chemically characterized, showing a nanoparticle with a mean diameter of 33 nm, crystalline nature and semiconductor behavior with a direct band gap of 2.2 eV. Moreover, the CuS nanoparticles exhibited localized surface plasmonic resonance in the near infrared range, with a high absorption band centered at 973 nm of wavelength, which allowed an increase in the temperature of the surrounding media under irradiation. Finally, the cytotoxicity of the CuS nanoparticles as well as their potential use as part of drug delivery platforms were investigated.

## 1. Introduction

Copper sulfide (CuS) nanoparticles have received significant attention during recent years due to their optoelectronic properties as well as their biocompatibility [1,2]. CuS nanoparticles exhibit *p*-type semiconductor behavior, with a high absorption coefficient, high electron mobility and low resistivity [3]. Although CuS is a *p*-type semiconductor, CuS nanostructures and thin films have also exhibited a metallic character at room temperature, associated with the CuS_3_ unit of the hexagonal crystalline structure, due to the moderately empty sulfur 3p bands [4], and even a superconductivity under 1.6 K [5]. Theoretical calculations have shown the low density of state between the valence band and conduction bands, which agrees with experimental data [6,7], and CuS nanoparticles have reached a conductivity of 550 S/cm [8]. On the other hand, they also demonstrated high catalytic and photocatalytic activities [9,10] and biocompatibility via in vivo and in vitro assays [11]. Many studies have demonstrated that CuS nanoparticles only exhibit toxic effects when they are used at a relative high concentration (higher than 100 µg/mL), and the expression of oxidated stress-genes is not related to their presence [12,13,14,15]. Moreover, they exhibit localized surface plasmon resonance (LSPR) in the near-infrared region [16,17]. This phenomenon is known as a coherent collective oscillation of electrons on the surface of nanoparticles, which arises when the frequency of an incident electromagnetic wave matches the natural frequency of a material, leading to a range of fascinating optical properties with potential applications in the fields of photocatalysis energy storage [18,19], solar cells [7] and biomedical [1,20].

The LSPR effect enables the CuS nanoparticles to selectively absorb and confine light energy in a localized manner, being able to convert that energy into heat, resulting in efficient photothermal agent conversion, which can increase the temperature of surrounding media [21,22]. The temperature increase could depend on different parameters, such as the nanoparticle concentrations and laser power density. The photothermal effect of CuS nanoparticles makes them suitable for various biomedical applications. One interesting application is the photothermal therapy used to alleviate arterial inflammation and restenosis, which significantly affects the long-term prognosis of endovascular treatment [23]. Studies demonstrated that CuS nanoparticles can induce localized hyperthermia upon near-infrared light irradiation, reducing the proliferation and migration of vascular smooth muscle cells and the expression of inflammatory cytokines [23]. Moreover, the photothermal effect of CuS nanoparticles can be also useful for magnetic resonance imaging, computed tomography, and photoacoustic imaging [2,24]. In addition, this effect can also be used for improving controlled drug delivery systems. CuS nanoparticles can be combined with another host material to increase the control of the drug released by employing NIR radiation. Conjugated CuS–chitosan nanocomposite showed efficient release of dopamine using NIR illumination [25]. Similarly, CuS–tetradecanol composites demonstrated the managed delivery of DOx [26]. In this context, nanoporous silicon matrices (nPSi) have been probed as efficient structures for drug delivery applications due to their high drug loading capacity, improved stability, and enhanced biocompatibility [27]. Moreover, their high surface area and tunable pore size allowed [28] them to be combined with other nanomaterials, such as metal nanoparticles, to improve the kinetic release of the drug [29].

The remarkable versatility of CuS nanoparticles in various industrial applications makes them highly attractive for cutting-edge research. However, conventional synthesis methods often involve chemical routes, leading to the generation of waste materials. Hydrothermal and solvothermal methods are the techniques most used to synthesize CuS nanoparticles. However, these techniques usually need the use of hazardous chemicals [30,31]. Other synthesis methods, such as co-precipitation, thermolysis or microwave-assisted solvothermal, also utilize risky chemicals or additional energy consumption [32,33,34,35]. To increase the sustainability of the fabrication process for nanoparticles, new approaches have been developed using industrial wastes as chemical sources [36]. An alternative and attractive approach to biorecover CuS nanoparticles involves recycling the precursor from drained mine waste waters using hydrogen sulfide acid, a colorless gas, produced through the metabolic process of sulfate-reducing bacteria (SRB) [37]. Sulfate contamination in water resources is a major environmental concern, especially in areas with significant industrial activity such as mining [38,39]. The most common method for sulfate removal involves chemical neutralization with lime, although this approach is not always effective in meeting the required standards and generates hazardous sludge requiring proper disposal [40]. As a more sustainable alternative, bioprocesses utilizing SRB have shown promising results in promoting sulfate’s reduction to hydrogen sulfide (H_2_S), which can be used to precipitate metal sulfides [41,42]. Bioprocesses that use SRB have great potential for recovering valuable metal sulfides from mining activities and other industrial processes [43]. This approach is sustainable, cost-effective, and environmentally friendly, making it an attractive alternative to conventional chemical synthesis methods for metal sulfide nanoparticles. 

In a previous work, our research group obtained CuS nanoparticles with high photocatalytic activity by using the biogenic H_2_S produced with a sulfidogenic bioreactor from real AMD [44]. By using an off-line system, biogenic H_2_S can react with Cu ions present in the AMD, forming CuS precipitates. Due to the solubility of CuS at a low pH, the only metal sulfide formed using H_2_S (g) is CuS. In addition, by controlling the H_2_S flux, the reaction can be tuned to obtain CuS nanoparticles. The aim of this study is to investigate the plasmonic and photothermal properties of CuS nanoparticles bioproduced using the same biotechnology method and to study their possible use in drug delivery applications. The obtained CuS nanoparticles were physico-chemically characterized and their LSPR, photothermal activity and cytotoxicity were studied. Finally, they were added to a porous silicon matrix and the potential controlled light-assisted drug delivery application was researched using caffeic acid as a drug model. Caffeic acid is phenolic acid that presents a wide range of biological and pharmacological activities, including antioxidant, anti-inflammatory, anticancer, and neuroprotective effects [45].

## 2. Results and Discussion

### 2.1. Physicochemical Characterization of CuS Nanoparticles

CuS nanoparticles were obtained after delivery of a controlled flux of H_2_S produced by the sulfidogenic bioreactor to an off-line vessel containing real Chilean AMD. The AMD was characterized by a copper concentration of 298 ppm, sulfate concentration of 3400 ppm and a pH of 3.8. Previous reports have demonstrated that under a flow rate of H_2_S of 100 mL∙min^−1^, more than 400 ppm of Cu ion can be removal from AMD after 60 min, biorecovering chemically pure CuS nanoparticles [44]. Given the relatively low copper concentration of the AMD, the CuS precipitates were recovered after delivery of the same H_2_S flux over just 60 min. 

Figure 1a shows a TEM image of the final CuS precipitates. Nanoparticles can be observed with a non-clear shape and a high tendency toward agglomeration. Figure 1b shows the histogram size of the nanoparticles, in which a mean diameter of 33 ± 9 nm could be estimated for nanoparticles with a spherical-like shape such as the one observed in the inset of Figure 1b. 

The crystalline structure of the CuS nanoparticles was characterized using the XRD technique (Figure 1c). The XRD pattern displayed prominent peaks at angles of 27.66° (101), 29.26° (102), 31.77° (103), 32.83° (006), 38.88° (105), 47.95° (110), 52.74° (108), 59.36° (116), and 74.06° (208). The obtained diffraction data were compared with the RRUFF database (code R060143.2), which confirmed a match with the covellite phase structure. The lattice parameters were determined to be a=b=3.6 $\dot{A}$ and  c=16.95 $\dot{A}$. The lattice parameters are in agreement with previous reports, where the nanoparticles were prepared using the chemical synthesis method [46,47,48]. The angle parameters are α=β=90° and γ=119.98°, providing further characterization of the crystalline structure. To determine the average nanocrystal size, the highest peak corresponding to the (110) plane was selected and calculated using the Debye–Scherrer equation:(1)$$\tau =\frac{K\lambda }{FWHM\cos\theta }$$
where τ is the average crystalline size (nm), K is a form factor set to 0.9, $\lambda =1.5406\;\dot{A}$ is the X-ray wavelength, *FWHM* is the full width at half-maximum of the diffraction peak (in radians) and θ is the Bragg angle of the diffraction peak (in radians). The calculated average crystalline size was 18.9 nm. Similar crystalline sizes were achieved in CuS nanoparticles synthesized via chemical methods such as the two-phase colloidal technique [48,49]. 

The UV–vis absorption spectrum of the CuS nanoparticles is depicted in Figure 1d, revealing semiconductor behavior. To determine the direct band gap, a Tauc plot was utilized. By plotting ${\left(\alpha hv\right)}^{2}\;$versus energy (eV) (inset of Figure 1d) and extrapolating the linear region to the horizontal axis, the direct band gap was determined to be 2.2 eV, as expected for this kind of nanoparticle, in which the band gaps can be tuned from 2.0 eV to 2.5 eV, depending on the structure and morphology of the nanoparticles [30,35,50].

### 2.2. Plasmonic Properties

To gain insights into the plasmonic properties of CuS nanoparticles, UV–vis–near-infrared spectroscopy analysis was performed. Figure 2a shows a prominent absorption band in the near-infrared range centered at 973 nm. This band indicates the strong absorbance of the nanoparticles within this wavelength range, resulting from the resonance of the free electrons on the nanoparticle surface. Furthermore, Figure 2b displays the absorption cross-section calculated using the BEM method for spherical CuS nanoparticles with different diameters: 24, 33 and 42 nm. The simulations show an absorption band in the same range as the experimental spectrum for all the diameters studied. The bigger the size of the nanoparticles, the higher the intensity of this band. The peak of the absorption cross-section is centered at 973 nm for nanoparticles with a diameter of 42 nm, and at 970 nm for those with diameters of 33 nm and 24 nm, in concordance with the experimental absorbance data, which also show a peak centered at approximately 973 nm.

To achieve deeper knowledge about the plasmon resonance of the CuS nanoparticles, the enhanced electric field within an illuminated nanoparticle was simulated. This calculation allowed us to analyze the impact of illumination on the electric field intensity near the nanoparticle. Figure 3 shows the enhanced electric fields of spherical CuS nanoparticles with different diameters 24 nm (Figure 3a), 33 nm (Figure 3b) and 42 nm (Figure 3c) under the illumination of an incident monochromatic wave of 970 nm. It can be observed that for all the sizes, the electric field intensity is enhanced on the surface of the nanoparticle. Moreover, the intensity of the electric field increases with the size of the nanoparticle, which is in agreement with the absorption cross-section simulations.

Figure 4 shows the effect of the enhanced electric field on the surface of a spherical CuS nanoparticle of 33 nm in diameter when it is illuminated under different monochromatic wavelengths: 920 nm, 970 nm and 1010 nm for Figure 4a, Figure 4b and Figure 4c, respectively, compared with the experimental absorption spectrum (Figure 4d). It is observed that, as the wavelength approaches the resonance peak, the enhancement of the electric field increases. Under an illumination of 920 nm (Figure 4a), in which the CuS nanoparticles do not absorb the light due to the absence of plasmon resonance in this range, the electric field enhancement is minimal. However, at the wavelength of 970 nm, corresponding to the plasmon resonance peak, the CuS nanoparticles exhibit significantly higher light absorption, leading to a substantial enhancement of the electric field (Figure 4b). Figure 4c also shows a black nanoparticle, the one illuminated with 1010 nm. This black nanoparticle, with just an electric field enhancement in its surface, is due to the BEM approach used here. By construction, the BEM is concerned with the electromagnetic response inside and outside of the nanoparticles, and when the plasmon resonance frequency is not matched, the electric field inside of the nanoparticle is much lower than the one on the surface due to the surface area being much bigger that the volume of the nanoparticles.

### 2.3. Photothermal Effect

Figure 5 shows the heating effect of water solution with and without CuS nanoparticles under irradiation with a monochromatic laser beam of 970 nm (0.1 W/cm^2^). It can be observed that the temperature increases from 12.5 °C to 17.8 °C after 10 min of illumination without CuS nanoparticles. However, the temperature reaches 27.1 °C with 2 mg/mL of CuS nanoparticles, demonstrating the photothermal effect of these nanostructures. 

Modeling the CuS nanoparticles with such as heat source (Equation (13)) and solving the heat equation, the expected solution (Equation (15)) is also plotted in Figure 5. Based on this solution, the temperature increase for the CuS nanoparticles is lower than the calculated one. However, in a previous report, Riedinger et al. obtained a similar result regarding the temperature increase, utilizing a comparable set-up with an NIR laser beam operating at 808 nm and 0.8 W/cm^2^ [51]. This finding suggests that calculations can be enhanced by either increasing the power per unit area of the laser beam or altering the wavelength of the plasmon resonance. It is crucial to emphasize that the shape and size of the nanoparticles are parameters that significantly influence the photothermal conversion process [52,53]. This result suggests that the low stability and the high tendency to agglomerate of these nanoparticles may dissipate the heat between the nanoparticles, making difficult the transference of heat to the surrounding medium. In order to achieve a more accurate model to predict the photothermal effects of these nanoparticles, further studies should be carried out, considering that heat dissipates between nanoparticles due to their tendency to agglomerate.

### 2.4. Cytotoxic Assays

The cytotoxic effect of CuS nanoparticles was determined in cultures of the MCF7 cell line incubated with increasing concentrations of CuS from 0 to 0.5 g/L for 6 h (Figure 6). It can be observed that no significant morphological changes are presented with the lower concentration of CuS nanoparticles (0.05 mg/mL) compared to the control (Figure 6a,b). Moreover, the cell viability is similar (Figure 6e). However, for concentrations higher than 0.1 mg/mL, clear cell damage can be appreciated (Figure 6c,d). In addition, the cell viability falls from 69% to 14% for concentrations of 0.1 and 0.5 mg/mL, respectively. Therefore, the cytotoxic effect of these CuS nanoparticles is dose-dependent, being innocuous for concentrations lower than 0.1 mg/mL. Similar effects have been reported using chemically synthesized CuS nanoparticles. Haiyan et al. reported that cell viability was compromised when using 0.1 g/L and higher after 24 h of incubation of NPs in cell culture [30]. Feng et al. also observed a decrease in viability at concentrations higher than 0.1 g/L when using different cell lines and a primary culture [54]. Furthermore, a reduction in cell viability was observed from concentrations of 0.12 g/L by Arshad et al. [32]. However, in the latter two studies, both observations were made at 24 and 48 h of exposure and with nanoparticles dissolved in vinyl pyrrolidine (PVP), and the decrease in cell viability was not abrupt, which could be explained by the presence of PVP. 

### 2.5. Drug Delivery Experiments

CuS nanoparticles with infrared plasmon resonance and photothermal activity could be used as a tool for biomedical applications such as drug delivery, since they can be activated by NIR light and control the kinetic drug release [55,56]. In order to study the use of these CuS nanoparticles in drug deliver applications, they were incorporated into functionalized nPSi matrices, which have been amply studied as efficient platforms for drug loading and release [27,28]. The nPSi–CuS samples were fabricated as shown in Figure 7. After electrochemical etching to form nPSi layers, the matrices were oxidized by H_2_O_2_ and functionalized with APTS. Finally, CuS nanoparticles were incorporated via capillary suption. 

To demonstrate the success of each step in the fabrication process for nPSi–CuS platforms, Figure 8 shows the FTIR spectrum at every step. The spectrum of the fresh nPSi layer shows two absorption peaks at 1085 and 615 cm^−1^ of wavenumber related to Si–O–Si and Si–Si vibration [57] The presence of a partially oxidized surface on this sample is due to the high reactivity of nPSi, which when in contact with the atmosphere prior to the analysis, could be partially oxidized. After the oxidation process (nPSi–Ox), the spectrum shows a more prominent and stronger peak at 1045 cm^−1^ and a stretching mode shoulder at 1178 cm^−1^. These signals correspond to the asymmetric stretching of the Si–O bonds in the Si–O–Si groups, confirming the success of the oxidation process [58]. The FTIR spectrum of nPSi–Ox functionalized with APTS (nPSi–Ox–APTS) shows three main absorption bands at 3093 cm^−1^, 1624 cm^−1^, and 1027 cm^−1^, which are associated with the N–H stretching band, NH_2_ bending vibrations, and Si–O–Si stretching band, respectively [59]. An additional weak signal was observed at 2936 cm^−1^, which could be assigned to the C–H bound due to asymmetrical and symmetrical stretching vibrations of CH_2_ groups [59]. Finally, after the incorporation of CuS nanoparticles (nPSi–CuS), the main absorption bands disappear. This fact can be associated with the high reflectance of the CuS nanoparticles at this wavelength range, which inhibits the absorption of the functional groups [60].

To study the morphology of the nPSi–CuS platform, SEM analysis was carried out (Figure 9). Figure 9a shows the cross-section of the nPSi–CuS platform, reveling that the nPSi structure had a thickness of 9.3 μm and the CuS nanoparticles were incorporated to a depth of 4.6 ± 0.2 microns from the surface of the nPSi, indicating the effective incorporation of the nanoparticles within the porous structure. Moreover, Figure 9b shows the surface of the nPSi–CuS platform, in which the presence of distributed CuS nanoparticles on the surface can be observed, indicating the successful incorporation of the nanoparticles also onto the surface.

The nPSi–CuS platforms were studied as controlled drug delivery systems using caffeic acid as a drug model. Caffeic acid is one of the most abundant drugs naturally available in a wide range of agricultural products, and its molecular structure is characterized by a phenylpropanoid (C6–C3) structure with a 3,4-dihydroxylated aromatic ring joined to a carboxylic acid (Appendix A) [61,62]. The drug loading capacity was 0.9 ± 0.1 µg/mm^2^, and the drug profile releases were performed in PBS. The experimental results of the fraction of the drug release in time (see Materials and Methods Section 3.5. Drug Delivery Test) are shown in Figure 10a. It can be observed that after 120 min, 80% of the loaded caffeic acid was delivered. To obtain deeper knowledge about the mechanisms that govern the kinetic release of the drug, the experimental data were fitted to different release models, including zero-order and first-order models, the Korsmeyer–Peppas model, the Higuchi model and the Hixson–Crowell model (Figure 10a). The main parameters of each fitting are shown in Table 1. Based on the $R_{adj}^{2}$ value, the release model that best suited the experimental data was the Korsmeyer–Peppas model, suggesting a Fickian diffusion process. In addition, since the release exponent *n* was smaller than 0.5, the erosion mechanism was negligible, pointing to the release process being purely diffusive.

To prove that the photothermal effect of the CuS nanoparticles due to plasmonic resonance in the NIR range can control the drug delivery profile, similar experiments were performed under irradiation with a 970 nm laser beam (0.1 W/cm^2^). Figure 10b shows the percentage of drug release from the nanostructure, both with and without NIR light illumination. When the sample is illuminated with NIR light, more than 80% of the drug is released during the first 5 min, being faster and higher drug release than obtained without NIR illumination. The advantage of using NIR light is that it has higher tissue penetration depth due to the minimal attenuation and refraction by endogenous chromophores and biomolecules [63]. This result confirms the usefulness of these CuS nanoparticles for photothermal biomedical applications.

To study the possible interaction between CuS nanoparticles and caffeic acid, FTIR measurements were performed on CuS loaded with a saturated solution of caffeic acid (Appendix A). The results do not conclude a direct interaction between both structures, and further studies should be carried out to deepen the knowledge about these mechanisms. It seems that the caffeic acid was encapsulated into the porous silicon matrix and the increase in the localized temperature due to the photothermal effects of the CuS nanoparticles allowed the rapid release of the drug out of the platform.

## 3. Materials and Methods

### 3.1. Materials

Acid mine drainage was recovered from a Chilean mine located in the V region. Glycerol, basic salt and trace element were purchased from Merk Millipore (Darmstadt, Germany). Moreover, *p*-type silicon wafers were purchased from University Wafer S.A. (Boca Raton, FL, USA). Hydrofluoric acid and ethanol for the synthesis of the porous silicon layer were purchased from Merck Millipore (Darmstadt, Germany). H_2_O_2_ (70%) and PBS was purchased from Merck Millipore (Darmstadt, Germany). 3-aminopropyl triethoxysilane was obtain from Sigma Aldrich (St. Louis, MO, USA), and caffeic acid were obtained from Sigma Aldrich (St. Louis, MO, USA). The MCF7 cell line was obtained from the ATCC (American Type Culture Collection) under code HTB-22.

### 3.2. Biosynthesis of CuS Nanoparticles

Copper sulfide nanoparticles were synthesized using the H_2_S gas stream produced by a sulfidogenic bioreactor of 2.3 L working volume (Fermac 200; Electrolab Biotech, Tewkesbury, UK), using glycerol as the electron donor and operating as an up-flow continuous biofilm at pH 3.5. The H_2_S produced by the bioreactor was removed in a stream of oxygen-free nitrogen (OFN) with a flow rate of 100 mL∙min^−1^, containing 0.1 (*v*/*v*) of H_2_S. The gas stream was delivered to an off-line vessel containing the real AMD. The solution was sparged by the gas stream for 60 min, mediating the precipitation of copper sulfide nanoparticles. The precipitates were collected by taking out the supernatant AMD solution and placing the CuS nanoparticles into a Falcon tube of 50 mL. Straight away, to wash the CuS nanoparticles, the Falcon tube was filled with distilled water at pH 3.3 and centrifugated at 3000 rpm for 30 min. Then, the supernatant was taken out and the process was repeated three times. Finally, the precipitates were dried in an oven at 30 °C for 24 h and stored for further use.

### 3.3. Synthesis of Hybrid Platform nPSi–CuS

The nPSi layers were formed via electrochemical etching of *p*-type silicon wafers (<100>, ρ = 0.001–0.005 Ω·cm, 500 µm of thickness) in HF:ethanol (1:2) solutions for 150 s, using a constant applied current density of 60 mA/cm^2^. The fresh nPSi layers were immersed in H_2_O_2_ solution for 24 h to obtained oxidized nPSi layers. Immediately, the oxidized nPSi layers were functionalized with amino groups by immersing them into APTS:ethanol (1:1000) solutions for 1 h. After that, the thin films were washed with ethanol and dried. Straight away, CuS nanoparticles were included via capillary suction, dropping 1 mL of 1 mg/mL of CuS aqueous solution drop by drop under orbital shaking (50 rpm). The CuS suspension was prepared by adding 50 mg of dried CuS nanoparticles into 50 mL of distilled water. The CuS suspension was sonicated at 50 W and 20 kHz (Q55; QSONICA, Newtown, CT, USA) before being dropped on the thin film. Finally, the nPSi–CuS hybrid platforms were washed with ethanol, dried and stored.

### 3.4. Characterization Techniques

The copper and sulfate concentrations of the AMD were measured via bicinchoninate and turbidimetric methods, respectively, using Hach (model DR 3900, Hach Company, Loveland, CO, USA) visual test kits (Hach CuVer 1 Copper Reagent Powder Pillows, Loveland, CO, USA, and Hach SulfurVer 4 Sulfate Reagent Powder Pillows, Loveland, CO, USA). The morphological characterization of the CuS nanoparticles was carried out using transmission electron microscopy (TEM) (1200 EXII model, JEOL, Tokyo, Japan). For that, 50 µL of an aqueous solution of CuS (1 mg/mL) was dropped onto a Cu TEM grid (grid size 200 mesh, Sigma Aldrich, St. Louis, MO, USA) and left to dry under room temperature conditions. The crystalline structure characterization of the CuS nanoparticles was carried out using Bruker D4 Endeavor X-ray diffraction equipment (XRD) with copper K_α_ radiation. The readings were taken within a 2θ range of 20 to 80 degrees, with a 0.02-degree step and a speed of 0.5 degrees per second. Then, 25 mg of CuS nanoparticles were placed on a silicon substrate and the reading area was set at 10 mm. The UV–vis absorption spectra were obtained using a BioTek (Winooski, VT, USA) spectrophotometer (EPOCH model). The measurement was performed by filling a well of a non-treated 96-well cell culture plate (Thermo Fisher Scientific Inc, Hillsboro, OR, USA) with 200 µL of an aqueous solution of CuS (1 mg/mL). The NIR absorption spectra measurements were carried out using a Varian spectrophotometer (Cary®50 model, Palo Alto, CA, USA) by filling a quart tray of 3 mL with an aqueous solution of CuS (1 mg/mL). The FTIR measurements were carried out in transmittance mode with a Jasco spectrophotometer (4600 model) in the range from 600 to 4000 cm^−1^. The morphological characterization of the nPSi–CuS platform was carried out via scanning electron microscopy (SEM) (JEOL, JSM 6010 PLUS/LA model). 

The photothermal assays were performed using an infrared laser (ALPHALAS, LDD1-1BT-LN, Goettingen, Germany). A well of 200 μL was filled with an aqueous solution of CuS nanoparticles (2 mg/mL) and the laser was set at a distance of 0.5 cm above the well. The temperature was registered at different time intervals with a thermocouple (VWR Traceable®, Radnor, PA, USA). As a control, the experiment was also performed with water without nanoparticles.

### 3.5. Drug Delivery Test

The nPSi–CuS samples were cut into pieces of 4 × 4 mm^2^ in area and loaded with caffeic acid by immersing them into a saturated aqueous solution (1 mM) for 24 h under orbital shaking (80 rpm). After that, the samples were washed with distilled water and placed into vials of 1 mL filled with PBS. At different time intervals, the supernatant solution was renewed and the caffeic acid concentration was measured via UV–vis spectroscopy at 310 nm [64]. The loading capacities of the nPSi–CuS samples were obtained by breaking up the loaded samples on the NaOH solutions, and the total loaded drug was characterized by measuring the supernatant via UV–vis spectroscopy. All the experiments were carried out in triplicate. 

To explore the drug release mechanism, the experimental profiles were fitted to different models, including the zero-order, first-order, Korsmeyer–Peppas, Higuchi and Hixson–Crowell models [65,66]. 

The zero-order equation is given by:(2)$$F=K_{0}t$$
where F is the fraction (%) of drug released in time *t*, specifically, *F* = *M*(*t*)/*M_∞_*, where *M*(*t*) as the amount of the drug released at time *t* and *M_∞_* is the final amount of drug at equilibrium (steady state). The value of *M_∞_* was 0.72 ± 0.1 µg/mm^2^ (82.88%) and it was reached at *t_∞_* = 240 min, and $K_{0}$ is the zero-order release kinetic constant.

The first-order equation is:(3)$$F=100\times \left(1-e^{-K_{1}t}\right)$$
where $K_{1}\;$is the first-order release kinetic constant.

The Korsmeyer–Peppas equation follows as:(4)$$F=K_{KP}t^{n}$$

$K_{KP}\;$ is the Korsmeyer–Peppas constant and n is the drug release parameter.

The Higuchi equation is:(5)$$F=K_{H}t^{1/2}$$

$K_{H}\;$ is the Higuchi release constant.

The Hixson–Crowell equation is:(6)$$F=100\cdot \left[1-{\left(1-K_{HC}t\right)}^{3}\right]$$

$K_{HC}\;$ is the Hixson–Crowell release constant.

### 3.6. Cell Culture and Cytotoxicity Assays

The cytotoxicity and cell viability assays were carried out using the breast adenocarcinoma line (MCF7). To keep the cells in culture, the culture medium DMEM (HyClone, Cytiva, Marlborough, PA, USA) supplemented with 10% fetal bovine serum (HyClone, Cytiva, Marlborough, PA, USA), 1% penicillin/streptomycin (Corning, Corning, NY, USA), and 1X glutamine (Gibco, Thermo Fisher) was used. To maintain the cells, the culture conditions were set to 37 °C and 5% CO_2_. The culture was expanded when it reached 80% confluence and the medium was changed every three days. To perform the cytotoxicity assays, the cells were seeded at a density of 5 × 10^4^ cells per well and settled overnight. The incubations with the CuS nanoparticles in complete DMEM were carried out at concentrations of 0, 0.05, 0.1, and 0.5 g/L for 6 h at 37 °C and 5% CO_2_. After that, the cells were trypsinized and counted with a Neubauer chamber using the 0.4% trypan blue exclusion method (Gibco, Thermo Fisher).

### 3.7. Simulations

#### 3.7.1. Localized Surface Plasmon Resonance

To gain a better understanding of the opto-electric properties of CuS nanoparticles, particularly focusing on the electromagnetic enhancement and the resulting heating process due to the LSPR, simulations were performed. For simplicity, the simulations were restricted to homogeneous, isotropic, non-magnetic materials. Additionally, a spherical-shaped nanoparticle was chosen, embedded in an aqueous medium (water). These simulations were essential for calculating the absorption cross-section, $\sigma_{Abs}$, quantity directly related to the heat generation mechanism within the nanoparticles. It is defined with respect to the absorbed power $P_{Abs}$ in dielectric or metallic objects during the scattering process and at the incident field intensity $S_{Inc}.$ The equation describing this relationship is as follows:(7)$$\sigma_{Abs}=P_{Abs}/S_{Inc}$$
where $S_{Inc}$ can be derived by considered the complex pointing vector of a propagating plane wave, that is: $S=\frac{1}{2}E\times H^{*}$, where $E=E_{0}e^{ik\cdot r}$, $H=H_{0}e^{ik\cdot r}$ and $H^{*}$ is the complex conjugate of the induced magnetic field H. $E_{0}$ and $H_{0}$ are complex vectors and **k** is the wave vector. The real amplitudes of the fields are related through $\left|H_{0}\right|=\sqrt{\epsilon_{M}}\left(\epsilon_{0}c\right)\left|E_{0}\right|$, which led to $S=S_{Inc}n$, where $S_{Inc}=\epsilon_{0}c\sqrt{\epsilon_{M}}{\left|E_{0}\right|}^{2}/2$, n is a real unit vector corresponding to the direction of the wave propagation, $\epsilon_{0}$ is the electric permittivity in free space, $\epsilon_{M}\;$is the electric permittivity of the medium and $\left|E_{0}\right|$ is the real amplitude of the incoming plane wave [67]. $P_{Abs}$ can be calculated via:(8)PAbs=−∫∫SRe12E×H*⋅nSr2dS
where *S* is the surface enclosing all the dielectric or metallic object and $n_{s}$ is the surface normal vector.

To find **H** and **E** in this study, the boundary elements method (BEM) developed by Garcia de Abajo and coworkers [68] was employed to perform simulations of the absorption cross-section and electric field enhancement of the CuS nanoparticles. Briefly, the BEM approach consists of expressing the three-dimensional dependence of the scalar and vector potentials, ϕ and A**,** respectively, in terms of the interface charges and currents via a set of surface integral equations described by the frequency-dependent local dielectric function $\epsilon \left(\omega \right)$ separated by sharp boundaries $\partial V_{j}$ [68] where the boundary is split into elements of finite size suitable for a numerical implementation. The scalar and vector potential are connected through the electric and magnetic fields as follows:(9)$$E=ikA-\nabla \phi ,\;\;\;\;\;\;H=\frac{1}{\mu }\nabla \times A$$
where k=ω/c is the wave vector. Equation (8) can be recast by using the Lorentz gauge ∇⋅A=ikϵμϕ into a set of two wave equations [69], which can be solved in terms of Green’s function:(10)$$\phi_{j}\left(r\right)=\phi_{j}^{e}\left(r\right)+\oint_{\partial V_{j}}G_{j}\left(\left|r-s\right|\right)\sigma_{j}\left(s\right)ds$$
(11)$$A_{j}\left(r\right)=A_{j}^{e}\left(r\right)+\oint_{\partial V_{j}}{G_{j\;}\left(\left|r-s\right|\right)h}_{j}\left(s\right)ds$$
where $\partial V_{j}$ is the boundary of the enclosed surface S. The scalar and vector potentials that characterize the external perturbation are given as:(12)$$\phi_{j}^{e}=\frac{1}{\epsilon_{j}\left(\omega \right)}\int dr^{\prime }G_{j}\left(\left|r-r^{\prime }\right|\right)\rho \left(r^{\prime }\right),\;\;A_{j}^{e}=\frac{\mu_{j}\left(\omega \right)}{c}\int dr^{\prime }{G_{j}\left(\left|r-r^{\prime }\right|\right)j}_{j}\left(r^{\prime }\right)$$
where $\sigma_{j}$ and $\;h_{j}$ are the charge and current distributions, and $G_{j}\left(r\right)=e^{ik_{j}r}/r$ is Green’s function of the wave equation. The charge and current distribution can be calculated by applying the appropriate boundary conditions for the equivalent boundary sources. In the context of the BEM method, the relative permeability is *µ* = 1, which is for a non-magnetic material. To obtain the values of the dielectric function ϵ(ω)=n(ω)+ik(ω) (complex and frequency-dependent) of the nanoparticles, an estimation was carried out using the Kramer–Kronig relationship, which correlates the absorption coefficient k with the absorption spectra, and the refractive index n [70]. The boundary condition can be imposed from the continuity of the scalar and vector potentials at the particle boundaries, together with the gauge condition and the dielectric displacement [68]

The simulations were conducted with three different spherical nanoparticles with diameters of 24, 33 and 42 nm. These diameters were selected based on the experimental average diameter of CuS nanoparticles, which is approximately 33 ± 9 nm. The chosen set-up for the simulations involved placing the nanoparticle at 5 nm away from a silicon (Si) surface. 

#### 3.7.2. Temperature Elevation of Nanoparticles via NIR Illumination

Plasmonic nanoparticles exhibit a heating effect upon illumination due to their strong light absorption in the NIR range. This heating is particularly significant in the plasmon resonance wavelength, creating a sharp temperature gradient at the interface between the nanostructure and the surrounding medium. This effect is described by the heat flow equation, which is dependent on time t and position r, as given by:(13)$$\nabla^{2}T\left(r,t\right)-\frac{1}{\kappa }\frac{\partial T\left(r,t\right)}{\partial t}=\frac{Q\left(r\right)}{k}$$
where $T\left(r,t\right)\;$is the local temperature, κ and  k are the thermal diffusivity and the thermal conductivity of the medium, respectively, and the function $Q\left(r\right)$ is the steady state heat source. Considering a planewave source propagating in the r direction, the steady state heat source is typically given by $Q\left(r\right)=I_{0}\mu_{e}e^{{-\mu }_{e}r}$, where $\mu_{e}$ is the effective attenuation coefficient (typically set to 0.9 [71]) and $I_{0}\;$is the plane wave intensity of the laser beam. However, assuming all the optical power is converted into heat and the nanoparticles are uniformly distributed in the medium, each nanoparticle will generate heat at a rate of $P=I_{0}\sigma_{Abs}$. With a number density of N particles per unit volume, the gap between them will be $d=N^{\left(-1/3\right)}.$ Therefore, the heat produced in the fluid per unit volume is:(14)$$Q=NP=NI_{0}\sigma_{Abs}=\frac{\sigma_{Abs}I_{0}}{d^{3}}$$

Equation (13) has an analytical solution, assuming that the nanoparticles are in a sphere of radius R and using the center temperature as our estimate from the nanoparticles. The heat flow equation was solved, assuming a number density of $N=10^{9}/cm^{3}$ in an aqueous solution of 200 μL and with a plane wave intensity of $0.1\;W/cm^{2}$, and the thermal diffusivity and thermal conductivity for the water are $\kappa =0.143\;{mm}^{2}\cdot s^{-1}$ and $k=0.6089\;W\cdot m^{-1}\cdot K^{-1}$, respectively. The solution for the temperature elevation is given by [72]:(15)$$T\left(z,t\right)=-\frac{K}{k\mu_{e}^{2}}\left[e^{-\mu_{e}z}\right]+\frac{2K}{\pi k}\int_{0}^{\infty }\frac{s\sin\left(sz\right)}{\mu_{e}^{2}+s^{2}}e^{-\kappa s^{2}\left(t-t_{0}\right)}\left\{\frac{1-e^{-\kappa s^{2}t_{0}}}{s^{2}}\right\}ds$$
where *t_0_* is the initial temperature.

## 4. Conclusions

The CuS nanoparticles recovered from acid mine drainage using the H_2_S of a sulfidogenic bioreactor exhibited local surface plasmon resonance in the near-infrared region. This effect allows for the photothermal activity of these nanoparticles, which can increase the temperature in a controlled way. In addition, they showed low toxicity with concentrations below to 0.1 g/L. Moreover, they could be used for drug delivery applications, showing a more controlled mode with the use of an infrared laser. Thus, this biosynthesis-based method for the recovery of CuS nanoparticles can open the way for their use in biomedical applications.

## Figures and Tables

**Figure 1 ijms-24-16489-f001:**
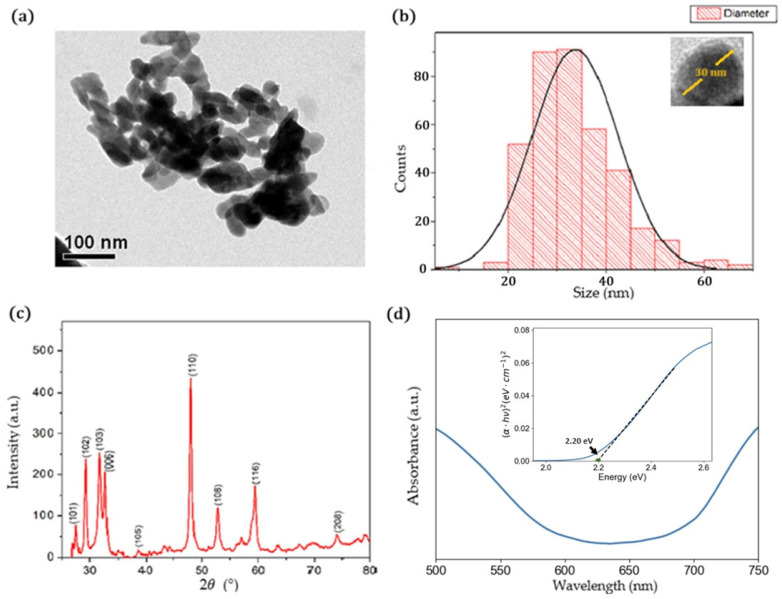
(**a**) TEM image of CuS nanoparticles. (**b**) Histogram size of CuS nanoparticles, inset showing a CuS nanoparticle with a spherical-like shape. (**c**) X-ray diffraction pattern of the CuS nanoparticle. (**d**) UV–vis absorption spectrum, inset showing the Tauc plot of the direct band gap.

**Figure 2 ijms-24-16489-f002:**
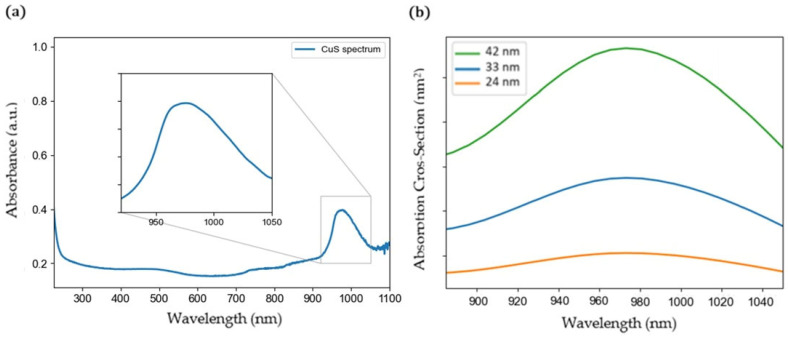
(**a**) Experimental UV–vis–NIR absorption spectrum of CuS nanoparticles. (**b**) Calculated absorption cross-section of CuS nanoparticles with different radii.

**Figure 3 ijms-24-16489-f003:**
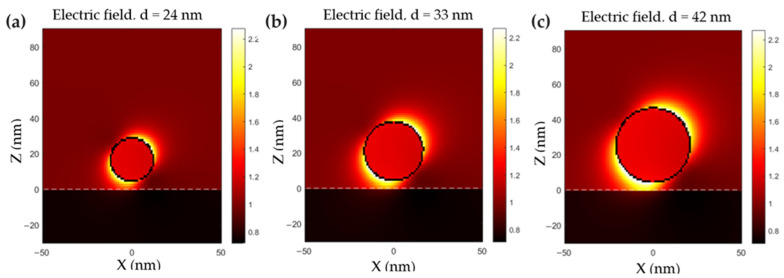
Calculated electric field enhancement (in Gaussian units) under illumination of 970 nm for spherical nanoparticles of (**a**) 24 nm, (**b**) 33 nm and (**c**) 42 nm in diameter.

**Figure 4 ijms-24-16489-f004:**
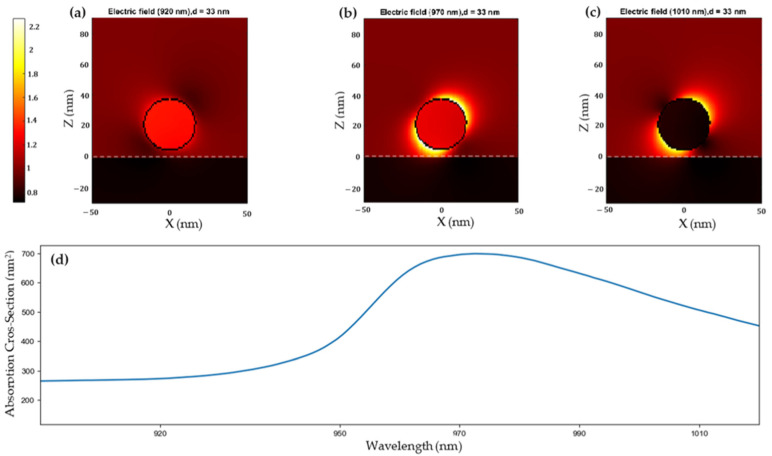
Simulations of the enhanced electromagnetic fields of a CuS nanoparticle with a diameter of 33 nm under NIR illumination of (**a**) 920, (**b**) 970 and (**c**) 1010 nm of wavelength, as compared to the experimental spectrum (**d**).

**Figure 5 ijms-24-16489-f005:**
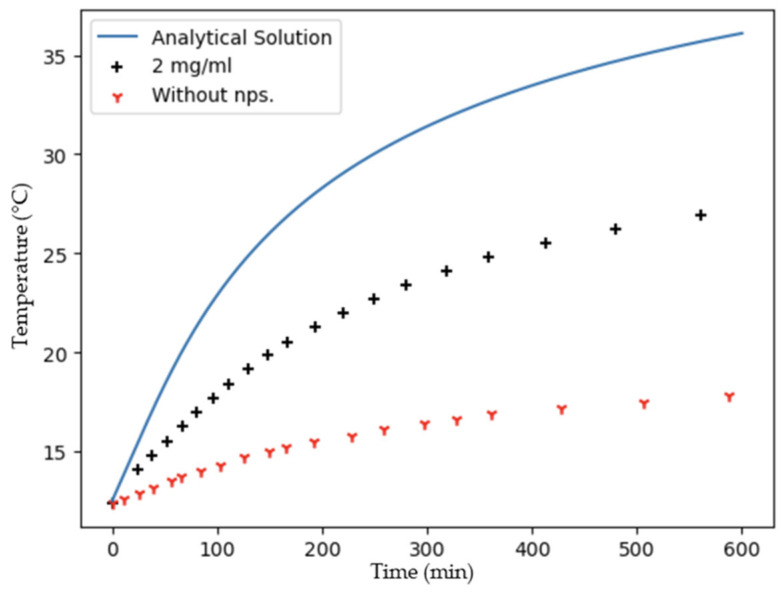
Experimental measurements of temperature versus time under near infra-red illumination with and without nanoparticles, as compared to the analytical solution.

**Figure 6 ijms-24-16489-f006:**
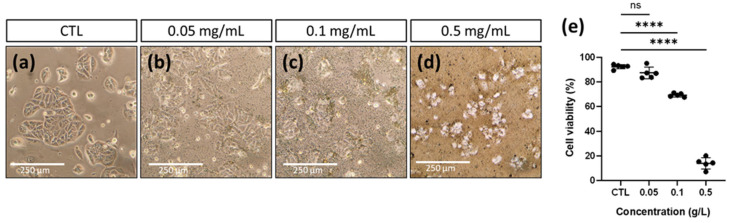
(**a**–**d**). Culture of the MCF7 cell line incubated with increasing concentrations of nanoparticles for 6 h. (**a**) Control, (**b**) 0.05 mg/mL, (**c**) 0.1 mg/mL, (**d**) 0.5 mg/mL and (**e**) cell viability. The percentage of cell viability mean ± SD; **** *p* < 0.05, ns means not significant, n = 6 from one experiment.

**Figure 7 ijms-24-16489-f007:**
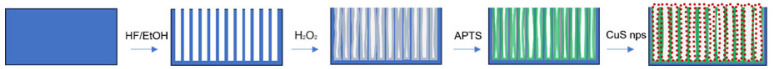
Scheme of nPSi–CuS functionalized process.

**Figure 8 ijms-24-16489-f008:**
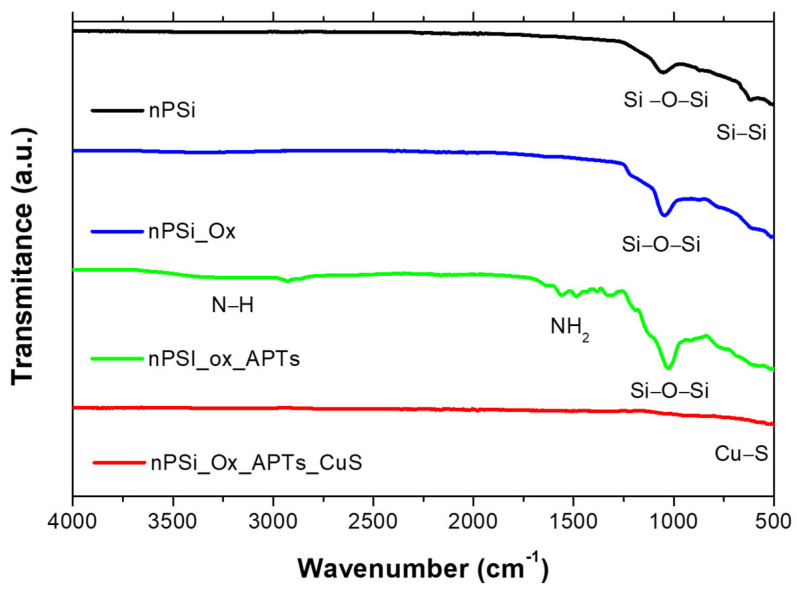
FTIR spectra at each step in the fabrication process for nPSi–CuS platforms. nPSi (fresh nPSi layer), nPSi_Ox (oxidized nPSi layer), nPSi_ox_APTs (functionalized nPSi layer), nPSi_Ox_APTs_CuS (Final nPSi-CuS platforms).

**Figure 9 ijms-24-16489-f009:**
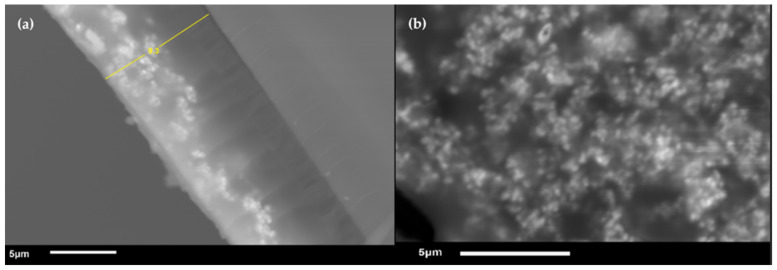
SEM images of (**a**) cross-section and (**b**) surface of nPSi–CuS platform.

**Figure 10 ijms-24-16489-f010:**
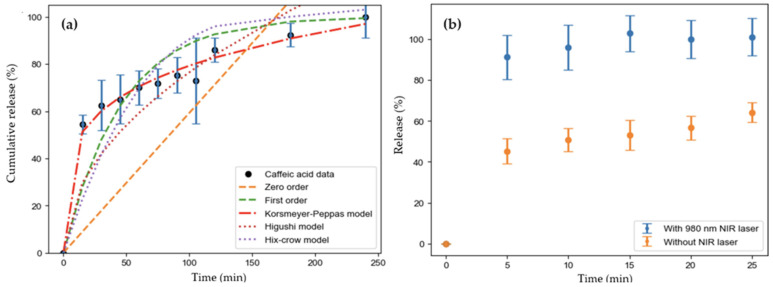
Experimental data concerning cumulative release as a function of time and their fitting to (**a**) kinetic release models and (**b**) caffeic acid release, with and without NIR light radiation.

**Table 1 ijms-24-16489-t001:** Kinetic models of caffeic acid release, where R^2^_adj_. is the adjusted regression coefficient; K_0_ is the zero-order release rate constant; K_1_ is the first-order release rate constant; K_KP_ is the Korsmeyer–Peppas release rate constant; *n* is the drug release exponent; K_H_ is the Higuchi constant, and K_HC_ is the Hixson–Crowell constant.

Model	Parameters		R^2^_adj_.
	*K*	*n*	
Zero order	0.59 ± 0.02		0.493
First order	0.021 ± 0.003		0.728
Korsmeyer–Peppas	27 ± 2	0.23 ± 0.01	0.985
Higuchi	7.6 ± 0.2		0.728
Hixson–Crowell	0.0054 ± 0.0006		0.673

## Data Availability

The data that support the findings of this study are available from the corresponding author upon reasonable request.

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
