# Peer review of "Plasmonic and Photothermal Effects of CuS Nanoparticles Biosynthesized from Acid Mine Drainage with Potential Drug Delivery Applications"

_ijms, 2023, doi:10.3390/ijms242216489_

Round 1

Reviewer 1 Report

Comments and Suggestions for Authors

The manuscript entitled “Plasmonic and photothermal effects of CuS nanoparticles biosynthesized from acid mine drainage with potential drug delivery applications” deals with  experimental and theoretical characterizations of CuS nanoparticles prepared from acid mine drainage (AMD} and H2S produced in a sulfidogenic bioreactor. The manuscript presents results of TEM, XRD, UV-VIS and NIR spectroscopies, photothermal effects, and cytotoxicity of the nanoparticles, as well as of their performance for the drug delivery. I appreciate experiments on drug delivery with/without the laser irradiation. There is merit to publish such a manuscript, however some details have to be added and explained in it as shown below. Thus,

1.       In Introduction, a reader would expect a brief description of standard chemical approaches for the preparation of CuS nanoparticles. The sentence in lines 59 and 60 is not correct, because the reference [21] is a review dealing with approaches for the treatment of different wastes. The approach described in this manuscript is an alternative to those described in this reference. References [22] and [29] are identical. Moreover, one can find one article on biosynthesized CuS nanoparticles in Science of The Total Environment 902 and thesis of H. Escobar-Sanchez on Google Scholar. As these sources are open ones, I strongly recommend commenting on their content in Introduction.

2.       In Part 2.2. it is necessary specify details how did you collect precipitate, washed it, and stored for further use. Did you dry the precipitate?

3.       In Part 2.3. specify please the thickness of the wafers used. How did you prepare the aqueous solution of CuS nanoparticles?

4.       In Part 2.4. please, describe how did you prepare samples for TEM, XRD, UV-VIS, NIR measurements.

5.       In Part 2.5. it is necessary to add some details on the caffeine measurements (wavelengths)? What does mean the content? Concentration or amount?   In Eqs. (1)-(5) it is necessary to think about meaning of F. Without doubts in Eqs (2) and (5), F=n/n¥=C/C¥ is a relative molar amount or molar concentration of released caffeine. The meaning of n¥  (C¥) is a steady-state molar amount (molar concentration) of released caffeine measured after a sufficiently long time (theoretically ¥). This molar amount (molar concentration) can be assumed as an adjustable parameter that improves the agreement of experimental data with fitted ones. How did you determine this parameter and F?  In the other formulas you can use either molar amount (molar concentration) or the relative molar amount (relative molar concentration).

6.       In Part 2.7.1., it is necessary briefly describe dielectrics and their shapes that you modelled. In line 182, the equation is not correct, and it is necessary to explain the meaning of E0 and introduce values eM for dielectrics considered. What is S in Eq. (7) and Vj in Eqs. (9) and (10)? What value of µ did you use?  Specify please, what method did you employ for the modelling and how did you choose appropriate boundary conditions?

7.       In Part 2.7.2., specify please the analytical solution of Eq. (12) (e.g., by using a reference) and values of k and k.

8.       In Part 3, it is necessary to add a short paragraph dealing with the nanoparticle preparation. In the paragraph you can deal with an average content of copper ions in AMD, the chemical purity of the prepared nanoparticles (e.g., by using some reference). It is also necessary to explain why you decided to precipitate CuS nanoparticles for 60 min. Longer precipitation times were used, elsewhere [28].

9.        In lines 241 and 253 some comparison with already published values of the lattice parameters and band gap, respectively, is necessary.

10.   In Line 314, compare please your results with published ones. It would be useful to propose how to change the model in Eq. (12) to obtain a better agreement between the experiment and calculations.

11.   In Fig. 10a, it seems that a steady-state value of F can be experimentally achieved after approximately 200 min. It is necessary to test the fittings of the first-order and Hixson-Crowell models including both K and n¥ adjustable parameters (see Comment 5).

12.   In Conclusions, the method of the precipitation of CuS nanoparticles is not new, because it has been already described. It would be sufficient to use e.g., “biosynthesis-based method”.

13.       I recommend a careful reading of a novel manuscript to avoid some unusual words such as capillary suption instead of capillary suction in line 113, precipitated instead of precipitate in line 104, orbital agitation instead of orbital shaking in line 114.

Comments on the Quality of English Language

Some comments are included into the Comments for Authors.

Author Response

REVIEWER 1:

The manuscript entitled “Plasmonic and photothermal effects of CuS nanoparticles biosynthesized from acid mine drainage with potential drug delivery applications” deals with  experimental and theoretical characterizations of CuS nanoparticles prepared from acid mine drainage (AMD} and H2S produced in a sulfidogenic bioreactor. The manuscript presents results of TEM, XRD, UV-VIS and NIR spectroscopies, photothermal effects, and cytotoxicity of the nanoparticles, as well as of their performance for the drug delivery. I appreciate experiments on drug delivery with/without the laser irradiation. There is merit to publish such a manuscript, however some details have to be added and explained in it as shown below. Thus,

  1. In Introduction, a reader would expect a brief description of standard chemical approaches for the preparation of CuS nanoparticles. The sentence in lines 59 and 60 is not correct, because the reference [21] is a review dealing with approaches for the treatment of different wastes. The approach described in this manuscript is an alternative to those described in this reference. References [22] and [29] are identical. Moreover, one can find one article on biosynthesized CuS nanoparticles in Science of The Total Environment 902 and thesis of H. Escobar-Sanchez on Google Scholar. As these sources are open ones, I strongly recommend commenting on their content in Introduction.

R: According to the reviewer’s suggestion, a brief description of chemical approaches for the fabrication of CuS nanoparticles has been incorporated in the introduction section at page 2 Line 70 “Hydrothermal and solvothermal methods are the most used techniques to synthesize CuS nanoparticles. However, these techniques usually need the use of hazardous chemicals [30,31]. Other synthesis methods such as co-precipitation, thermolysis or micro-wave-assited solvothermal, also utilize risky chemical or additional energy consumption [32–35]. To increase the sustainability in the fabrication process of nanoparticles, new ap-proaches have been developed using industrial wastes as chemical sources [36]”

Moreover, new references have been also included:

  1. Yadav, S.; Shrivas, K.; Bajpai, P.K. Role of Precursors in Controlling the Size, Shape and Morphology in the Syn-thesis of Copper Sulfide Nanoparticles and Their Application for Fluorescence Detection. J Alloys Compd 2019, 772, 579–592, doi:10.1016/j.jallcom.2018.08.132.
  2. Huang, Z.; Wang, L.; Wu, H.; Hu, H.; Lin, H.; Qin, L.; Li, Q. Shape-Controlled Synthesis of CuS as a Fenton-like Photocatalyst with High Catalytic Performance and Stability. J Alloys Compd 2022, 896, 163045, doi:10.1016/j.jallcom.2021.163045.
  3. Arshad, M.; Wang, Z.; Nasir, J.A.; Amador, E.; Jin, M.; Li, H.; Chen, Z.; Rehman, Z. ur; Chen, W. Single Source Precursor Synthesized CuS Nanoparticles for NIR Phototherapy of Cancer and Photodegradation of Organic Carcinogen. J Photochem Photobiol B 2021, 214, 112084, doi:10.1016/j.jphotobiol.2020.112084.
  4. Motaung, M.P.; Onwudiwe, D.C.; Wei, L.; Lou, C. CuS, In2S3 and CuInS2 Nanoparticles by Microwave-Assisted Solvothermal Route and Their Electrochemical Studies. Journal of Physics and Chemistry of Solids 2022, 160, 110319, doi:10.1016/j.jpcs.2021.110319.
  5. Tetyana, P.; Mphuthi, N.; Jijana, A.N.; Moloto, N.; Shumbula, P.M.; Skepu, A.; Vilakazi, L.S.; Sikhwivhilu, L. Syn-thesis, Characterization, and Electrochemical Evaluation of Copper Sulfide Nanoparticles and Their Application for Non-Enzymatic Glucose Detection in Blood Samples. Nanomaterials 2023, 13, 481, doi:10.3390/nano13030481.
  6. Pejjai, B.; Reddivari, M.; Kotte, T.R.R. Phase Controllable Synthesis of CuS Nanoparticles by Chemical Co-Precipitation Method: Effect of Copper Precursors on the Properties of CuS. Mater Chem Phys 2020, 239, 122030, doi:10.1016/j.matchemphys.2019.122030.
  7. Abdelbasir, S.M.; McCourt, K.M.; Lee, C.M.; Vanegas, D.C. Waste-Derived Nanoparticles: Synthesis Approaches, Environmental Applications, and Sustainability Considerations. Front Chem 2020, 8, doi:10.3389/fchem.2020.00782.

On the other hand, previous reference 29 have been changed to “Nancucheo, I.; Segura, A.; Hernández, P.; Canales, C.; Benito, N.; Arranz, A.; Romero-Sáez, M.; Recio-Sánchez, G. Bio-Recovery of CuS Nanoparticles from the Treatment of Acid Mine Drainage with Potential Photocatalytic and Antibacterial Applications. Science of The Total Environment 2023, 902, 166194, doi:10.1016/j.scitotenv.2023.166194.”, and more details about the work done in this reference have been added at page 2 line 93:  “By using an off-line system, biogenic H2S can react with Cu ions present into the AMD, forming CuS precipitates. Due to the solubility of CuS at low pH, the only metal sulfide formed using H2S (g) is CuS. In addition, by controlling the H2S flux, the reaction can be tunned to obtain CuS nanoparticles.”

  1. In Part 2.2. it is necessary specify details how did you collect precipitate, washed it, and stored for further use. Did you dry the precipitate?

R: As the reviewer suggests, more details about the collection, wash and store of  the precipitates were added to the text in section 3.2. “Biosynthesis of CuS nanoparticles” at page 11 line 346 : “The precipitates were collected by taking out the supernatant AMD solution and placing the CuS nanoparticles into a Falcon tube of 50 mL. Straightaway, to wash the CuS nano-particles, the Falcon tube was filled with distilled water at pH 3.3 and centrifugated at 3000 rpm for 30 minutes. Then, the supernatant was taken out and the process was re-peated three times. Finally, the precipitates were dried in an oven at 30 °C for 24 hours and stored for further use.”

  1. In Part 2.3. specify please the thickness of the wafers used. How did you prepare the aqueous solution of CuS nanoparticles?

R: The thickness of the Si wafers was 500 µm. This was added to the manuscript at page 11 line 354 . Moreover, details of the preparation of the aqueous solution of CuS nanoparticles were included at page 11 line 361 “The CuS suspension was prepared by adding 50 mg of dried CuS nanoparticles into 50 mL of distilled water. The CuS suspension was sonicated at 50 W and 20 kHz (Q55; QSONICA, Newtown, USA) before to drop on the thin film”

  1. In Part 2.4. please, describe how did you prepare samples for TEM, XRD, UV-VIS, NIR measurements.

R: Details of the preparation of the samples for TEM, XRD, UV-VIS and NIR measurements have been added in section 3.4 Characterization techniques at page 12 line 370 “The morphological characterization of CuS nanoparticles was carried out by trans-mission electron microscopy (TEM) (JEOL, 1200 EXII model). For that, 50 µL of an aqueous solution of CuS (1 mg/mL) was dropped on a Cu TEM grid (grid size 200 mesh, Sigma Al-drich, St. Louis, MO, USA) and left to dry at room temperature conditions. The crystalline structure characterization of CuS nanoparticles was carried out using the Bruker D4 En-deavor X-ray diffraction equipment (XRD) with copper Kα radiation. The readings were taken within a 2q range of 20 to 80 degrees, with a 0.02-degree step and a speed of 0.5 de-grees per second. 25 mg of CuS nanoparticles were placed on a silicon substrate and the reading area was set at 10mm. UV-vis absorption spectra were obtained with a BioTek spectrophotometer (EPOCH model). The measurement was performed by filling a well of a non-treated 96 well cell culture plate (ThermoFisher Scientific Inc, Hillsboro, USA) with 200 µL of an aqueous solution of CuS (1 mg/mL). NIR absorption spectra measurements were carried out with a Varian Spectrophotometer (Cary®50 model) by filling a quart tray of 3 mL with an aqueous solution of CuS (1 mg/mL).”

  1. In Part 2.5. it is necessary to add some details on the caffeine measurements (wavelengths)?

R: The wavelength detail about the measurements of caffeic acid was included at page 12 line 397 “At different time intervals, the supernatant solution was renewed and the caffeic acid concentration was measured by UV-vis spectroscopy at 310 nm [65].”

In addition, a new reference was also added:

  1. Guzmán-Oyarzo, D.; Plaza, T.; Recio-Sánchez, G.; Abdalla, D.S.P.; Salazar, L.A.; Hernández-Montelongo, J. Use of NPSi-ΒCD Composite Microparticles for the Controlled Release of Caffeic Acid and Pinocembrin, Two Main Polyphenolic Compounds Found in a Chilean Propolis. Pharmaceutics 2019, 11, 289, doi:10.3390/pharmaceutics11060289.

What does mean the content? Concentration or amount?   

R: Thank you for your comment. We agree with the reviewer that the word “content” could induce a misunderstanding. We measured the fraction of the drug released by its concentration in the supernatant.

In Eqs. (1)-(5) it is necessary to think about meaning of F. Without doubts in Eqs (2) and (5), F=n/n¥=C/C¥ is a relative molar amount or molar concentration of released caffeine. The meaning of n¥  (C¥) is a steady-state molar amount (molar concentration) of released caffeine measured after a sufficiently long time (theoretically ¥). This molar amount (molar concentration) can be assumed as an adjustable parameter that improves the agreement of experimental data with fitted ones. How did you determine this parameter and F?  In the other formulas you can use either molar amount (molar concentration) or the relative molar amount (relative molar concentration).

R: Thank you for your comment. We agree that F value is not enough detailed in the manuscript. In all models, F is the fraction (%) of drug released in time t (doi: 10.1208/s12248-010-9185-1.). We have changed that in text at page 12 line 407:  “F is the fraction (%) of drug released in time t, specifically, F = M(t) / M, where M(t) as the amount of the drug released at time t and M is the final amount of drug at the equilibrium (steady-state). The value of M was 0.72 ± 0.1 µg/mm2 (80%), and was reached at t = 240 min.”

Moreover, a new reference has been included to support this information:

  1. Zhang, Y.; Huo, M.; Zhou, J.; Zou, A.; Li, W.; Yao, C.; Xie, S. DDSolver: An Add-In Program for Modeling and Comparison of Drug Dissolution Profiles. AAPS J 2010, 12, 263–271, doi:10.1208/s12248-010-9185-1.

  1. In Part 2.7.1., it is necessary briefly describe dielectrics and their shapes that you modelled. In line 182, the equation is not correct, and it is necessary to explain the meaning of E0 and introduce values eMfor dielectrics considered. What is S in Eq. (7) and ¶Vj in Eqs. (9) and (10)?

R: As the reviewer suggests, the dielectric and the shape were describe at page 12 line 441“ For simplicity, the simulations were restricted to homogeneous, isotropic, non-magnetic materials. Additionally, a spherical-shaped nanoparticle was chosen, embedded in an aqueous medium (water).”

Regardless to S in equation 7, more details about its meaning were added at page 14 line 449:  “where  can be derived by considered the complex pointing vector of a propagating plane wave, this is:   , where H* is the complex conjugate of the induced magnetic field. The real amplitude of the fields are related through  this lead to , where , n is a real unit vector corresponding to the direction of the wave propagation, ϵ_0 is the electric permittivity in free space,   is the electric permittivity of the medium and |E_0 | is the real amplitude of the incoming plane wave”.

On the other hand, abut the meaning of , it was clarified in page 14 line 472: “where  are the boundary of the enclosed surface s”.

What value of µ did you use? 

R: The value of µ  used in this model was 1, according to non-magnetic material. This was added to the text at page 15 line 477: “In the context of the BEM method the relative permeability is equal to µ=1, which is for a non-magnetic material.”

Specify please, what method did you employ for the modelling and how did you choose appropriate boundary conditions?

R: The choose of the boundary conditions and the method employed for the simulations was also describe in details in page  15 line 478: “To obtain the values of the dielectric function ϵ(ω)=n(ω)+ik(ω) (complex and frequency dependent) of the nanoparticles, it was carried out an estimation of it by using the Kramer-Kronig relationship, which correlates the absorption coefficient k with the absorption spectra, and the refractive index n [71]. The boundary condition can be imposed from the continuity of the scalar and vector potentials at the particle boundaries, together whit gauge condition and the dielectric displacement [69]

The simulations were conducted with three different spherical nanoparticles with diameters of 24 nm, 33 nm, and 42 nm. These diameters were selected based on the experimental average diameter of CuS nanoparticles, which is approximately 33 ± 9 nm. The chosen setup for the simulations involved placing the nanoparticle at 5 nm away from a silicon (Si) surface.”

  1. In Part 2.7.2., specify please the analytical solution of Eq. (12) (e.g., by using a reference) and values of k and k.

R:  The values of the thermal diffusity and the thermal conductivity used in this work were  0.143 mm2·s-1 and  0.6089 W·m-1·K-1 , respectively. These value were added to the text at page 15 line 508 with the analytical solution of equation 12 “the thermal diffusivity and thermal conductivity for the water are κ= 0.143 mm^2 \s and k= 0.6089 W·m−1·K−1 respectively. The solution for the temperature elevation is given by [73]:

where t0 is the initial temperature.”

Moreover, a new reference was also included to verify the analytical solution:

  1. Norton, S.J.; Vo-Dinh, T. Nanoparticle-Mediated Heating: A Theoretical Study for Photothermal Treatment and Photo Immunotherapy. In; 2021; pp. 89–114.

  1. In Part 3, it is necessary to add a short paragraph dealing with the nanoparticle preparation. In the paragraph you can deal with an average content of copper ions in AMD, the chemical purity of the prepared nanoparticles (e.g., by using some reference). It is also necessary to explain why you decided to precipitate CuS nanoparticles for 60 min. Longer precipitation times were used, elsewhere [28].

R: The copper content of the AMD was 298 ppm. In a previous report, under a flux rate of H2S of 100 mL/min, more than 400 ppm of copper ions can be removal from the AMD and bioprecipitated in the form of pure CuS nanoparticles (https://doi.org/10.1016/j.scitotenv.2023.166194). Thus, since the relative low concentration of Cu ions in this AMD, we decided to synthetize CuS nanoparticles only for 60 minutes. This has been added into the manuscript in a short paragraph at the beginning of the Result and Discussion section, as the reviewer suggests, at page 3 line 107 “CuS nanoparticles were obtained after delivered a controlled flux of H2S produced from the sulfidogenic bioreactor to an off-line vessel containing a real Chilean AMD. The AMD was characterized by a copper concentration of 298 ppm, sulfate concentration of 3400 ppm and a pH of 3.8. Previously reports have demonstrated that under a flow rate of H2S of 100 ml∙min-1, more than 400 ppm of Cu ion can be removal from an AMD after 60 minutes, biorecovering chemical purely CuS nanoparticles [48]. Since the relative low copper concentration of the AMD, the CuS precipitates were recovered after delivered the same H2S flux during 60 minutes.”

Moreover, details about the measurements of copper and sulfate content in the AMD were also added in the materials and method section at page 11 line 366 “Copper and sulfate concentration of AMD were measured by bicinchoninate and tur-bidimetric methods, respectively, using Hach (model DR 3900, Hach company) visual test kits (Hach CuVer 1 Copper Reagent Powder Pillows and Hach SulfurVer 4 Sulfate Rea-gent Powder Pillows ).”

  1. In lines 241 and 253 some comparison with already published values of the lattice parameters and band gap, respectively, is necessary.

R: As the reviewer suggests, the lattice parameters and band gap results obtained for these CuS nanoparticles were compared to other published works. The comparison of lattice parameters has been added to the text in page 3 at line 125: “The lattice parameters are in agreement with previous reports, where the nanoparti-cles were prepared through chemical synthesis method [46–48].”

Moreover, the comparison of the crystalline size was included at page 3 line 134. “Similar crystalline sizes were achieved in CuS nanoparticles synthesized by chemical methods such as two phases colloidal technique [48,49].”

And new references have been also included:

  1. Mohamed, M.B.; Abdel‐Kader, M.H.; Almarashi, J.Q.M. Role of Cu/S Ratio and Mg Doping on Modification of Structural and Optical Characteristics of Nano CuS. Int J Appl Ceram Technol 2020, 17, 832–840, doi:10.1111/ijac.13337.
  2. Deb, S.; Kalita, P.K. Green Synthesis of Copper Sulfide (CuS) Nanostructures for Heterojunction Diode Applica-tions. Journal of Materials Science: Materials in Electronics 2021, 32, 24125–24137, doi:10.1007/s10854-021-06879-2.
  3. Hussein, O.N.; AL-Jawad, S.M.H.; Imran, N.J. Efficient Antibacterial Activity Enhancement in Fe/Mn Co-Doped CuS Nanoflowers and Nanosponges. Bulletin of Materials Science 2023, 46, 139, doi:10.1007/s12034-023-02964-w.
  4. Al-Jawad, S.M.H.; Taha, A.A.; Muhsen, M.M. Preparation and Characterization of CuS Nanoparticles Prepared by Two-Phase Colloidal Method. J Phys Conf Ser 2021, 1795, 012053, doi:10.1088/1742-6596/1795/1/012053.

On the other hand, the band gap value was also compared to other reported values for CuS nanoparticles. The comparison was added to the text at page 3 line 140 “the direct band gap was determined to be 2.2 eV, as expected for this kind of nanoparticles, in which the band gaps can be tunned form 2.0 eV to 2.5 eV, depending on the structure and morphology of the nanoparticles  [35, 50, 51]”.

  1. In Line 314, compare please your results with published ones. It would be useful to propose how to change the model in Eq. (12) to obtain a better agreement between the experiment and calculations.

R: The photothermal results were compared to other published ones. Moreover, a proposal to modify the heat transfer model considering the heat dissipates between nanoparticles was also added at page 7 line 205 “Modeling the CuS nanoparticles such as heat source (Equation (13)) and solving the heat equation, the expected solution (Equation (15)) is also plotted in Figure 5. Based on this solution, the temperature increase for CuS nanoparticles is lower than the calculated one. However, in a previous report, Riedinger et al. obtained a similar result regarding the temperature increase, utilizing a comparable set-up with a NIR laser beam operating at 808 nm and 0.8 W/cm² [52]. This finding suggests that calculations can be enhanced by either increasing the power per unit area of the laser beam or altering the wavelength of the plasmon resonance. It is crucial to emphasize that the shape and size of the nanoparticles are parameters that significantly influence the photothermal conversion process [53,54]. This result suggests that the low stability and the high tendency to agglomerate of these nanoparticles may dissipated the heat between the nanoparticles making difficult the transference of heat to the surrounding medium. In order to achieve a more accurate model to predict photothermal effects of these nanoparticles, further studies should be carried out con-sidering the heat dissipates between nanoparticles due to their tendency to agglomerate.”

In addition, three new references have been included:

  1. Riedinger, A.; Avellini, T.; Curcio, A.; Asti, M.; Xie, Y.; Tu, R.; Marras, S.; Lorenzoni, A.; Rubagotti, S.; Iori, M.; et al. Post-Synthesis Incorporation of 64 Cu in CuS Nanocrystals to Radiolabel Photothermal Probes: A Feasible Ap-proach for Clinics. J Am Chem Soc 2015, 137, 15145–15151, doi:10.1021/jacs.5b07973.
  2. Kim, M.; Lee, J.; Nam, J. Plasmonic Photothermal Nanoparticles for Biomedical Applications. Advanced Science 2019, 6, doi:10.1002/advs.201900471.
  3. Wu, H.; Shen, L.; Tian, L.; Zhao, F.; Yin, J.; Shao, Y. Excitation of Quantum-Sized (CuS)n Clusters for NIR-II Pho-tothermal Application: Insights from Experiment and DFT Calculation. J Alloys Compd 2023, 941, 169005, doi:10.1016/j.jallcom.2023.169005.

  1. In Fig. 10a, it seems that a steady-state value of F can be experimentally achieved after approximately 200 min. It is necessary to test the fittings of the first-order and Hixson-Crowell models including both K and n¥ adjustable parameters (see Comment 5).

R: We have recalculated the fitting of all the models using t = 240 min. Figure 10a and Table 1 have been updated with the new fitting. However, the release model which better suited with the experimental data is still the Korsmeyer-Peppas model with a release exponent lower than 0,5, suggesting a purely Fickian diffusive process.

  1. In Conclusions, the method of the precipitation of CuS nanoparticles is not new, because it has been already described. It would be sufficient to use e.g., “biosynthesis-based method”.

R: Thank you for your comment. The last sentence of the conclusion part has been modified to: “Thus, this biosynthesis-based method of recovery CuS nanoparticles can open the way for their use in biomedical applications.”

  1. I recommend a careful reading of a novel manuscript to avoid some unusual words such as capillary suption instead of capillary suction in line 113, precipitated instead of precipitate in line 104, orbital agitation instead of orbital shaking in line 114.

R: Thank very much for your comment. We have carefully revised the new version of the manuscript and some English typos have been checked.

Reviewer 2 Report

Comments and Suggestions for Authors

The manuscript by Sanchez et al. describes a new way of fabrication of CuS nanoparticles. The research is very well planned and conducted, most of the conclusions are sound and well-described.

There are minor corrections necessary:

1. The issue of biocompatilibity of CuS nanoparticles should be presented in detail - two references, not to top journals, is not sufficent, especially if the issues of biocompatibility is essential for the paper. Please develop this section more.

2. Are really CuS nanoparticles so conductive - at the single nnaoparticle level, or as an assembly - please give more details to the introduction.

3. Reuno Unido should be "United Kingdon" I suppose.

4. Inset in Fig. 1d is barely visible - in all figures axil labels should be larger.

5. The molecular structure of caffeic acid should be shown and its interactino with the surface of CuS should be described. Bonding mode should be suggested (and verified experimentally, if possible, or at least computational (dft) evidence for bond formation should be given).

Comments on the Quality of English Language

Minor corrections necessary.

Author Response

The manuscript by Sanchez et al. describes a new way of fabrication of CuS nanoparticles. The research is very well planned and conducted, most of the conclusions are sound and well-described.

There are minor corrections necessary:

  1. The issue of biocompatilibity of CuS nanoparticles should be presented in detail - two references, not to top journals, is not sufficent, especially if the issues of biocompatibility is essential for the paper. Please develop this section more.

R: Thank you for your comment. We have added more details about the biocompatibility of CuS nanoparticles in the introduction section at page 1 line 35 “On the other hand, they also demonstrated high catalytic and photocatalytic activities [9,10], and biocompatibility in vivo and in vitro assays [11]. Many studies have been demonstrated that CuS nanoparticles only exhibit toxic effects when they are used at rela-tive high concentration (higher than 100 µg/mL), and the expression of oxidated stress-genes are not related by their presence [12–15]”.

New references have been also included:

  1. Li, Y.; Yang, Z.; Jalil, A.T.; Saleh, M.M.; Wu, B. In Vivo and In Vitro Biocompatibility Study of CuS Nanoparticles: Photosensitizer for Glioblastoma Photothermal Therapy. Appl Biochem Biotechnol 2023, 195, 4084–4095, doi:10.1007/s12010-023-04313-3.
  2. Li, X.; Yuan, H.J.; Tian, X.M.; Tang, J.; Liu, L.F.; Liu, F.Y. Biocompatible Copper Sulfide–Based Nanocomposites for Artery Interventional Chemo-Photothermal Therapy of Orthotropic Hepatocellular Carcinoma. Mater Today Bio 2021, 12, 100128, doi:10.1016/j.mtbio.2021.100128.
  3. Saona, L.A.; Campo-Giraldo, J.L.; Anziani-Ostuni, G.; Órdenes-Aenishanslins, N.; Venegas, F.A.; Giordana, M.F.; Díaz, C.; Isaacs, M.; Bravo, D.; Pérez-Donoso, J.M. Cysteine-Mediated Green Synthesis of Copper Sulphide Nano-particles: Biocompatibility Studies and Characterization as Counter Electrodes. Nanomaterials 2022, 12, 3194, doi:10.3390/nano12183194.
  4. Chakraborty, S.; Prasad, R.; Pandey, P.K.; Khan, A.; Jain, N.K.; Jones, E.V.; Srivastava, R.; Misra, S.K. Doxorubicin Encapsulated Hollow Self-Assembled CuS Nanoparticles Clusters for Bio-Responsive Chemo-Photo Therapy. Mater Lett 2022, 327, 133017, doi:10.1016/j.matlet.2022.133017.
  5. Peng, S.; He, Y.; Er, M.; Sheng, Y.; Gu, Y.; Chen, H. Biocompatible CuS-Based Nanoplatforms for Efficient Photo-thermal Therapy and Chemotherapy in Vivo. Biomater Sci 2017, 5, 475–484, doi:10.1039/C6BM00626D.
  6. Are really CuS nanoparticles so conductive - at the single nnaoparticle level, or as an assembly - please give more details to the introduction.

R: As the reviewer suggests, more details about the conductivity of the CuS nanoparticles have been included in the introduction section in page 1 at line 29: .” CuS nanoparticles exhibit p-type semiconductor behavior with high absorption coeffi-cient, high electron mobility and low resistivity [3]. Although CuS is a p-type semicon-ductor, CuS nanostructures have also exhibited a metallic character at room temperature associated to CuS3 unit of the hexagonal crystalline structure, due to the moderately empty sulfur 3p bands [4], and even a superconductivity under 1.6 K [5]. Theoretical calculations have shown low density of state between the valence band and conduction bands which agree with experimental data [6,7], and CuS nanoparticles have reached a conductivity of 550 S/cm [8].”

In addition, new references have been also added:

  1. Mageshwari, K.; Mali, S.S.; Hemalatha, T.; Sathyamoorthy, R.; Patil, P.S. Low Temperature Growth of CuS Na-noparticles by Reflux Condensation Method. Progress in Solid State Chemistry 2011, 39, 108–113, doi:10.1016/j.progsolidstchem.2011.10.003.
  2. Casaca, A.; Lopes, E.B.; Gonçalves, A.P.; Almeida, M. Electrical Transport Properties of CuS Single Crystals. Jour-nal of Physics: Condensed Matter 2012, 24, 015701, doi:10.1088/0953-8984/24/1/015701.
  3. Gainov, R.R.; Dooglav, A. V.; Pen’kov, I.N.; Mukhamedshin, I.R.; Mozgova, N.N.; Evlampiev, I.A.; Bryzgalov, I.A. Phase Transition and Anomalous Electronic Behavior in the Layered Superconductor CuS Probed by NQR. Phys Rev B 2009, 79, 075115, doi:10.1103/PhysRevB.79.075115.
  4. Morales-García, A.; Soares, A.L.; Dos Santos, E.C.; de Abreu, H.A.; Duarte, H.A. First-Principles Calculations and Electron Density Topological Analysis of Covellite (CuS). J Phys Chem A 2014, 118, 5823–5831, doi:10.1021/jp4114706.
  5. Tirado, J.; Roldán-Carmona, C.; Muñoz-Guerrero, F.A.; Bonilla-Arboleda, G.; Ralaiarisoa, M.; Grancini, G.; Queloz, V.I.E.; Koch, N.; Nazeeruddin, M.K.; Jaramillo, F. Copper Sulfide Nanoparticles as Hole-Transporting-Material in a Fully-Inorganic Blocking Layers n-i-p Perovskite Solar Cells: Application and Working Insights. Appl Surf Sci 2019, 478, 607–614, doi:10.1016/j.apsusc.2019.01.289.
  6. Singh, N.; Taunk, M. Structural, Optical, and Electrical Studies of Sonochemically Synthesized CuS Nanoparti-cles. Semiconductors 2020, 54, 1016–1022, doi:10.1134/S1063782620090262.
  7. Reuno Unido should be "United Kingdon" I suppose.

R: Thank you for your comment. The typo was checked.

  1. Inset in Fig. 1d is barely visible - in all figures axil labels should be larger.

R: We have modified the label of all the figures

  1. The molecular structure of caffeic acid should be shown and its interactino with the surface of CuS should be described. Bonding mode should be suggested (and verified experimentally, if possible, or at least computational (dft) evidence for bond formation should be given).

R: As the reviewer suggests, more details about the molecular structure of the caffeic acid were added to the manuscript at page 9 line 288: “ Caffeic acid is one of the most abundance drug which is naturally available in a wide range of agricultural products and its molecular structure is characterized by a phe-nylpropanoid (C6-C3) structure with a 3,4-dihydroxylated aromatic ring joined to a car-boxylic acid (Figure S1) [62,63]”.

In addition, two  references were also included:

  1. Belay, A.; Kim, H.K.; Hwang, Y. Probing the Interaction of Caffeic Acid with ZnO Nanoparticles. Luminescence 2016, 31, 654–659, doi:10.1002/bio.3007.
  2. Espíndola, K.M.M.; Ferreira, R.G.; Narvaez, L.E.M.; Silva Rosario, A.C.R.; da Silva, A.H.M.; Silva, A.G.B.; Vieira, A.P.O.; Monteiro, M.C. Chemical and Pharmacological Aspects of Caffeic Acid and Its Activity in Hepatocarci-noma. Front Oncol 2019, 9, doi:10.3389/fonc.2019.00541.

The molecular structure of the caffeic acid was included as supplementary material in Figure S1.

On the other hand, the interaction between the CuS nanoparticles and caffeic acid was investigated with FTIR measurements and the results were incorporated as supplementary material (Figure S2). However, the experimental results did not provide any clues about the interaction between both structures and further studies should be done to get deeper knowledge about their interaction. The authors suggest that the caffeic acid was encapsulated into the porous silicon matrix without direct interaction with the CuS nanoparticles. These suggestions were incorporated into the text at page 10 line 312: “To study the possible interaction between CuS nanoparticles and caffeic acid, FTIR measurements were performed on CuS loaded with a saturated solution of caffeic acid (see supplementary material, Figure S2). The results do not conclude a direct interaction be-tween both structures and further studies should be carried out to deep the knowledge about these mechanisms. It seems that the caffeic acid was encapsulated into the porous silicon matrix and the increase of the localized temperature due to the photothermal effects of the CuS nanoparticles allowed the rapid release of the drug out of the platform.”

Round 2

Reviewer 1 Report

Comments and Suggestions for Authors

Still, I have comments to the revised manuscript entitled Plasmonic and photothermal effects of CuS nanoparticles biosynthesized from acid mine drainage with potential drug delivery applications“

11.       It is strange that you have numbered parts "Materials and Methods“, and "Results and discussion“ as #3, and #2, respectively. Usually, the part on materials and methods  comes before the part on results and discussion. I strongly recommend to change the order of these parts in a novel manuscript.

22.       I agree with changes in paragraph 3.5. on drug delivery tests. However, the application of formulas shown in this paragraph on measured experimental data in Figs. 10a and 10b and Tab. 1 (Page 10 of this manuscript) is not correct. You write that F= M/M∞ with M∞≈80%. In such a case the experimental data in Figs. 10 should be higher and approach a value of 100% at times around 240 min, provided that "Cumulative release [%]"=F.  Instead, they reach a value of about 80%  which may indicate that M instead of F is used in Figs. 10. Then, formulas (3) and (6) cannot be correctly fitted into such data. Check and correct the fits, once more.

33.       I accept the other modifications made in the previous manuscript.

Author Response

REVIEWER 1

Still, I have comments to the revised manuscript entitled “Plasmonic and photothermal effects of CuS nanoparticles biosynthesized from acid mine drainage with potential drug delivery applications“

  1. It is strange that you have numbered parts "Materials and Methods“, and "Results and discussion“ as #3, and #2, respectively. Usually, the part on materials and methods  comes before the part on results and discussion. I strongly recommend to change the order of these parts in a novel manuscript.

R: We agree with the reviewer that it seems strange that the materials and methods section appears after the results. However, we made those changes according to the requirements of the journal since the editor asked for this modification.

  1. I agree with changes in paragraph 3.5. on drug delivery tests. However, the application of formulas shown in this paragraph on measured experimental data in Figs. 10a and 10b and Tab. 1 (Page 10 of this manuscript) is not correct. You write that F= M/M∞ with M∞≈80%. In such a case the experimental data in Figs. 10 should be higher and approach a value of 100% at times around 240 min, provided that "Cumulative release [%]"=F.  Instead, they reach a value of about 80%  which may indicate that M instead of F is used in Figs. 10. Then, formulas (3) and (6) cannot be correctly fitted into such data. Check and correct the fits, once more.

R: Thank you very much for your comment. We agree that Figures 10a and 10b must be modified to be in agreement with the proposal kinetic release models. In this way, we have modified both figures considering that F= M/M∞ with M∞=82,88%, and the kinetic models have been fitted once again to the experimental data. Then, the values of the Table 1 have been also updated. However, the model which better described the release of the drug is still the Korsmeyer-Peppas model with a release exponent n smaller than 0.5. To clarify these modifications, we have incorporated these changes into the text at page 9 line 292 “Experimental results of the fraction of drug release in time (see Material and Method sub-section 3.5. Drug delivery test) are shown in Figure 10a. It can be observed that after 120 minutes the 80% of the loaded caffeic acid was delivered”

      Moreover, we modify the value M∞=82,88% in order to give a more accurate value in page 12 line 410.

  1. I accept the other modifications made in the previous manuscript.

R: Thank you very much for all your comment which gave valuable improvements to our paper.

Round 3

Reviewer 1 Report

Comments and Suggestions for Authors

I recommend to publish this manuscript.